# Lemnaceae as Novel Crop Candidates for CO_2_ Sequestration and Additional Applications

**DOI:** 10.3390/plants12173090

**Published:** 2023-08-28

**Authors:** Marina López-Pozo, William W. Adams, Barbara Demmig-Adams

**Affiliations:** 1Department of Plant Biology & Ecology, University of the Basque Country, 48940 Leioa, Spain; 2Department of Ecology and Evolutionary Biology, University of Colorado, Boulder, CO 80309, USA

**Keywords:** duckweed, carbon dioxide, nitrogen, microbiome

## Abstract

Atmospheric carbon dioxide (CO_2_) is projected to be twice as high as the pre-industrial level by 2050. This review briefly highlights key responses of terrestrial plants to elevated CO_2_ and compares these with the responses of aquatic floating plants of the family Lemnaceae (duckweeds). Duckweeds are efficient at removing CO_2_ from the atmosphere, which we discuss in the context of their exceptionally high growth rates and capacity for starch storage in green tissue. In contrast to cultivation of terrestrial crops, duckweeds do not contribute to CO_2_ release from soils. We briefly review how this potential for contributions to stabilizing atmospheric CO_2_ levels is paired with multiple additional applications and services of duckweeds. These additional roles include wastewater phytoremediation, feedstock for biofuel production, and superior nutritional quality (for humans and livestock), while requiring minimal space and input of light and fertilizer. We, furthermore, elaborate on other environmental factors, such as nutrient availability, light supply, and the presence of a microbiome, that impact the response of duckweed to elevated CO_2_. Under a combination of elevated CO_2_ with low nutrient availability and moderate light supply, duckweeds’ microbiome helps maintain CO_2_ sequestration and relative growth rate. When incident light intensity increases (in the presence of elevated CO_2_), the microbiome minimizes negative feedback on photosynthesis from increased sugar accumulation. In addition, duckweed shows a clear propensity for absorption of ammonium over nitrate, accepting ammonium from their endogenous N_2_-fixing *Rhizobium* symbionts, and production of large amounts of vegetative storage protein. Finally, cultivation of duckweed could be further optimized using hydroponic vertical farms where nutrients and water are recirculated, saving both resources, space, and energy to produce high-value products.

## 1. Overview: Lemnaceae for CO_2_ Mitigation Coupled with Additional Attractive Features

Since the industrial revolution, human activity has resulted in increased levels of various atmospheric pollutants (including greenhouse gases), with atmospheric carbon dioxide (CO_2_) projected to be twice as high as the pre-industrial revolution level by 2050 [1,2]. As detailed below, duckweeds are efficient at removing CO_2_ from the atmosphere. In addition to CO_2_ mitigation, duckweed can provide food (for humans and livestock) of high nutritional quality at high yields with minimal input of light, space, and fertilizer supply (Figure 1). Moreover (Figure 1), duckweed can produce feedstock in the form of starch and lipid for production of biofuels [3,4] and can also serve in wastewater recycling and freshwater reclamation through the removal of nitrogen- and phosphate-based fertilizer run-off and other waste products [5].

Duckweed is also particularly suitable for hydroponic culture, which avoids CO_2_ release associated with soil tilling [6]. Under natural conditions, duckweed coverage has, furthermore, been shown to reduce or eliminate CO_2_ emissions from ponds [7].

### 1.1. Harvest Index

Unlike terrestrial crops, the entire duckweed plant is edible, which in turn reduces (or eliminates) post-harvest waste products. Much or all the plant (in rootless duckweeds) consists of one or several small green organs termed fronds. Figure 2 illustrates that duckweed fronds are unusual in simultaneously acting as the plant’s carbohydrate-source tissue (taking in light and CO_2_ as direct inputs into photosynthesis to produce sugars) as well as its major carbohydrate-sink tissue (because non-green/non-photosynthetic tissues are virtually absent). In contrast, land plants must invest in substantial non-photosynthetic support structures, including a root system, systems of long-distance vascular transport between roots and shoot, and structural support that allows leaves and stems to be displayed upright and withstand wind and a dry atmosphere. On the other hand, duckweed floating on water has minimal to no need for long-distance transport of nutrients or for structural support. As will be elaborated further in Section 5 below, the green frond is also chiefly responsible for nutrient uptake, especially under ample nutrient supply. Only under low nutrient supply do plants primarily rely on fine roots (with a superior surface-to-volume ratio) for nutrient uptake (except in species without roots).

### 1.2. The Plant Microbiome’s Multiple Roles

This section provides a brief background on plant-microbiome interactions based largely on the existing literature on terrestrial plants. The microbiome, in addition to maintaining or improving growth, is also responsible for other processes relevant to plant health and nutrition [8]. The microbiome contributes to bioavailability and plant absorption of nutrients (Figure 2), as well as production of various beneficial metabolites [9,10]. Specifically, the microbiome performs biological fixation of atmospheric nitrogen [11] and solubilization of phosphorus [12,13], potassium [10], and iron [14], as well as sulfur oxidation [15]. In addition, members of the microbiome synthesize compounds with structures and functions similar to those of plant hormones, as well as affect the regulation and synthesis of phytohormones produced by the plant itself [16]. The plant microbiome can thus influence plant growth and development [17] and nutrient uptake [18,19]. Additional impacts of the plant microbiome include induction of biotic defense [20], such as induced systemic resistance against pests, insects, and pathogenic fungi [21,22], and plant resistance to biotic stress [23]. Some members of the microbiome can also compete with pathogens for limiting nutrients, inhibit their growth, produce antibiotic compounds, or dissolve fungal cell walls [24,25]. Through these many impacts, the plant microbiome also modulates the plant responses to elevated CO_2_, which include phytohormone synthesis and signaling [26], as well as altered carbon allocation pathways [27], growth [28], senescence [29], and defenses [30]. In turn, the metabolic activity of the microbiome is supported by energy-rich carbohydrates supplied by the plant [31].

### 1.3. Growth Rate, Sink Strength, and Response to CO_2_

This section briefly summarizes literature on how plant growth and carbohydrate storage capacity affect the response to elevated CO_2_ across plant species and places the exceptionally high growth rate of Lemnaceae into this context. A plant’s sink strength, i.e., its capacity to either utilize or store photosynthetically produced sugar, plays a role in determining plant response to elevated CO_2_ [32,33,34]. Plants with a high sink strength tend to exhibit sustained new growth, storage, and/or reproductive output under elevated CO_2_ and otherwise favorable conditions (see Section 2). In contrast, plants with a low sink strength typically not only exhibit no sustained increases in photosynthetic productivity but can instead experience metabolic disruption [34] and accelerate senescence [35] under elevated CO2. Lemnaceae possess the highest relative growth rates among plants [36], which may be related to the fact that aquatic floating plants reinvest their photosynthetic income predominantly into additional productive infrastructure (additional photosynthetic tissue) while having minimal need to invest in non-productive (non-photosynthetic) support structures (see [37]). The resulting rapid frond growth, paired with a considerable starch-storage capacity of the frond itself [4,38], constitutes the major sinks within the plant, with carbohydrate consumption by the plant microbiome (Figure 2) acting as an additional sink [34,39,40] and serving to maintain high plant sink strength. Future research should address the capacity of duckweed to increase the allocation of any surplus sugar underutilized to polysaccharide synthesis (starch [32,41,42] but also cell wall material [43]). Maintenance of new growth under conditions of high carbohydrate supply is supported by matching input of mineral nutrients for an appropriate carbon-to-nitrogen ratio. The interaction of CO_2_ and nutrient supply is addressed for the case of duckweed (Section 2 below).

## 2. Context-Dependent Response of Duckweed to Elevated CO_2_: Role of Nutrients, Light, and Microbiome

This section briefly reviews the response of Lemnaceae to elevated CO_2_ in the context of different environmental conditions. 

### 2.1. Effect of Nutrient Supply and Plant Microbiome under Moderate Light Supply

Under ample nutrient supply (Figure 3a), elevated CO_2_ did not inhibit growth (as relative growth rate, RGR, of frond area expansion) in *Lemna minor*, irrespective of whether the plant microbiome was absent (sanitized plants) or present (inoculated plants). In contrast, RGR was inhibited by elevated CO_2_ in plants transferred to low nutrient supply when the microbiome was absent but not when the microbiome was present (Figure 3b). Plant-microbiome interaction thus supported *L. minor*’s performance under elevated CO_2_ in combination with limiting nutrient supply. It should be noted that this was a snapshot in time upon transfer of duckweed from high to low nutrient supply and that responses may change after internal nitrogen sources [storage protein] become further depleted.

The fact that elevated CO_2_ did not increase RGR in *L. minor* under ample nutrient supply suggests that RGR was already maximal under these conditions, allowing no further stimulation by elevated CO_2_. These fronds were grown in a nutrient medium containing both nitrate and ammonium. The following section addresses the additional effect of high light (in combination with elevated CO_2_) and raises the possibility that duckweed responses to both elevated CO_2_ and inoculation may vary depending on the specific form of nitrogen supplied (nitrate versus ammonium).

### 2.2. Response of Inoculated Lemna Plants under High Light and Ample Nutrient Supply

Figure 4a illustrates that RGR was unaffected by elevated CO_2_ even in combination with continuous high light (700 µmol photons m^−2^ s^−1^) input in inoculated *L. minor* plants (with their microbiome) growing with ample nutrient supply. Such a combination of high inputs from both light and CO_2_ can trigger detrimental effects in some plants because of extreme surplus sugar accumulation and resulting metabolic disruption [34]. The ability of duckweed to maintain its high RGR under such circumstances (Figure 4a) is consistent with early reports for duckweed by other authors (see, e.g., [45,46]) that elevated CO_2_ had no effect on RGR.

A somewhat different response of duckweed to elevated CO_2_ was reported by Toyama et al. [47], who grew plants in nutrient-replete nitrate-only medium (without ammonium) and reported that plant inoculation with beneficial microbes caused RGR to increase from ~0.36 to ~0.55 with concomitant changes in plant nitrogen metabolism and amino acid profile. Future research should compare responses side-by-side for duckweed grown in ammonium-containing versus ammonium-free media. Figure 4 compares responses under high light supply (Figure 4a,c,e,g,i) with those under moderate light supply (Figure 4b,d,f,h,j), all in nutrient-replete medium. Area RGR was just below 0.5 under 200 µmol photons m^−2^ s^−1^ (Figure 4b) and just above 0.5 under 700 µmol photons m^−2^ s^−1^ in either ambient or elevated CO_2_ in ammonium-containing medium (Figure 4a). Section 5 provides further detail on the preference of aquatic floating plants for ammonium.

For the sets of plants with their microbiome shown in Figure 4, dry biomass accumulation was unaffected by elevated CO_2_ under continuous high growth light intensity (700 µmol photons m^−2^ s^−1^) but was stimulated somewhat (from just below to just above half of that in very high light) under moderate light of 200 µmol photons m^−2^ s^−1^. This finding is consistent with the notion that elevated CO_2_ can have a stimulatory effect when plant productivity is not already saturated. One would consequently predict a subsequent downregulation of photosynthesis (affecting various chloroplast constituents) by elevated CO_2_ under the condition (very high light) where productivity is already saturated but not under the condition (lower light) where productivity can still increase. 

Pioneering work by Gale et al. [48] demonstrated this effect for duckweed—with an initial increase in photosynthesis rate upon increasing CO_2_ quickly, giving rise to a pronounced decrease in photosynthesis rate under high growth light intensity with a long photoperiod but not under low growth light intensity and/or a short photoperiod. The data in Figure 4 illustrates the results of this secondary downregulation of photosynthesis. Total chlorophyll content was downregulated in plants exposed to elevated versus ambient CO_2_ under 700 µmol photons m^−2^ s^−1^ with more surplus sugar, but not in plants growing under 200 µmol photons m^−2^ s^−1^ with less surplus sugar (Figure 4e,f). This photosynthetic downregulation but unimpeded growth under elevated CO_2_ reflects the general ability of plants (including duckweed) to support the same growth rate with a lesser investment in photosynthetic machinery in the presence of elevated CO_2_ by virtue of acclimatory downsizing of the photosynthetic apparatus to counteract surplus sugar production [34,37,49]. This downsizing of the photosynthetic apparatus includes downregulation of chlorophyll in the plants grown under continuous high light in the presence of elevated CO_2_ (Figure 4g) as well as downregulation of carotenoids (Figure 4h) bound to the same light-harvesting protein complexes as chlorophyll (see [37]). Such effects, as documented in Figure 4 for duckweed, are also observed in other plants. Because these carotenoids serve as important human micronutrients [50], their decline constitutes a cost to plant nutritional quality for the human (or livestock) consumer [51,52]. However, as Figure 4 illustrates, this cost can be prevented by growing plants under a lower light intensity (Figure 4g,h). 

Overall frond protein content was also lower under elevated CO_2_ in the presence of continuous high light (Figure 4j), albeit to a lesser extent than chlorophyll or carotenoid levels. This finding is likely unique to duckweed and is consistent with the role of Ribulose 1,5-bisphosphate Carboxylase/Oxygenase (RuBisCO), the predominant protein in fronds, not only in photosynthesis but also as a vegetative storage protein in duckweed [53] that is maintained at high levels across a wide range of growth light environments [54,55]. This welcome finding thus illustrates the unique ability of duckweed to maintain high protein content under elevated CO_2_. Moreover, Figure 4 illustrates that any negative effects of elevated CO_2_ on both protein and carotenoid micronutrient content can be prevented by growing plants under a lower light supply (Figure 4j) with ample nutrient supply. It is noteworthy that when the same relatively low light supply was paired with limiting nutrient supply, downregulation of carotenoid content was observed in sanitized plants [39,40]. In contrast, plants inoculated with their microbiome exhibited less or no such downregulation [39].

## 3. Multiple Advantages of Growing Duckweed under Low Light Intensity

This section compares key features of plant responses to light quality in land plants with those of Lemnaceae. As stated above, plant carotenoid content—with respect to the important human micronutrients lutein and provitamin A (β-carotene)—is highest under low growth light intensity in duckweed [38,56] as well as other plants. However, unlike terrestrial plants, duckweed can be grown under low light without a decline in RGR and with a maximally high protein content per dry mass [55]. In contrast to land plants, *Lemna* thus grew similarly fast, with the same light- and CO_2_-saturated photosynthetic capacity and as much protein per frond area (and more per dry biomass), under low versus high growth light intensity [56,57]. Such a high maximal photosynthetic capacity even in low-light-grown plants is consistent with duckweeds’ high RuBisCO content [55,58]. Overall, these findings indicate that the accumulation of desirable products relative to the required light energy input is most favorable in duckweed grown under low light intensity [56].

Moreover, unlike land plants with multiple layers of leaves, duckweeds have no fixed plant tiers and possess relatively thin leaves, resulting in minimal self-shading and the ability to saturate the whole plant even at low levels of incident light intensity. This scenario may also explain the fact that duckweeds accumulate exceptionally large amounts of antioxidants that serve as essential (= must be acquired from the diet) human micronutrients with antioxidative and anti-inflammatory effects (antioxidant vitamins A, C, E, carotenoids, and phenolics) [37,53,55,59,60]. In contrast, land plants have lower average levels of antioxidants per leaf dry mass, with relatively low antioxidant levels in the lower versus upper portions of individual leaves [61,62] as well as lower versus upper leaves in a tiered plant canopy [61]. In addition, land plants downsize their pools of various antioxidants and other essential human micronutrients under low growth light intensity more strongly than duckweed [54]. This difference between duckweeds and land plants, along with duckweed’s high RGR under low light [54], is advantageous for energy-efficient cultivation of duckweed in controlled environments with relatively low light supply (see also Section 7 and Section 8).

## 4. Light Quality Effects

The upper versus lower portions of leaves, as well as whole upper versus lower leaves, of most land plants absorb most of the blue and red fractions of incident sunlight and dissipate much of this as heat (via a photoprotective process [63]) at peak irradiance in natural settings, even in fast-growing plants (e.g., [64]). In tiered land plants, only little blue and red light penetrates down to the lower layers of leaves or canopies. In those lower levels, green and far-red portions of the solar spectrum thus represent most of the available energy [65]. These lower layers may require additional support from photosynthate supplied by the upper layers of the leaf or plant. Plants with such tiered structures may benefit from inclusion of wavelengths other than blue and red in their growth light supply. On the other hand, duckweeds (without fixed tiered layers and relatively thin fronds) may reach photosynthetic saturation with just the blue and red portions of the light spectrum, with little to no additional stimulation by the green and far-red regions of the solar spectrum.

## 5. Duckweed Preference for Ammonium versus Nitrate

Because ammonium is the most common form of nitrogenous waste produced by heterotrophic aquatic organisms, it is not surprising that aquatic duckweeds are highly efficient at nitrogen uptake in the form of ammonium (Figure 5a,b) and prefer ammonium over nitrate [66]. *Lemna minor*, transferred from ample to low nutrient supply, preferentially removed ammonium from the medium and only started drawing down the nitrate pool after the ammonium pool had been depleted (Figure 5d). Over the first two days following transfer from ample to limiting nutrient supply (Figure 5a,c; phase 1), there was no draw-down of nitrate and the draw-down of ammonium was much less pronounced compared to the next several days (Figure 5b,d, phase 2). This result suggests that new fronds growing during phase 1 were supported by internal nitrogen stores (vegetative storage protein). For the relatively low light intensities used here, a longer photoperiod led to more pronounced uptake of nitrogen compounds than a shorter photoperiod (Figure 5). A similar scenario, with nitrate uptake commencing only after ammonium had been consumed, was also seen in the duckweed genus *Landoltia* [67,68]. Duckweeds thus have a high fertilizer-use efficiency and capacity for phytoremediation [5] such as removal of fertilizer runoff from freshwater bodies and fertilizer retention in, e.g., rice paddies [69].

In closed photobioreactors, increased CO_2_ supply enhanced removal of nitrate and phosphate from the medium by duckweed [4], along with increasing RGR and starch content, indicating that higher nutrient removal efficiency was likely due to a higher growth rate under elevated CO_2_. In duckweeds, unlike in land plants, fronds and roots are both active in nutrient uptake, and both organs are more efficient at taking up ammonium compared to nitrate [70,71,72]. Whereas frond activity accounted for the majority of nutrient update under high nutrient supply, roots were responsible for most of both ammonium and nitrate uptake at low nutrient supply [70]. The ability to take up high levels of ammonium without experiencing the toxicity effects commonly observed in land plants [73,74] is related to duckweeds’ ability to rapidly convert ammonium to protein via an expanded glutamine oxoglutarate aminotransferase (involved in synthesis of glutamate from glutamine and α-ketoglutarate) gene family [75] and storage of large quantities of protein throughout the plant [76]. In this respect, duckweeds are like legumes—land plants capable of using ammonium and storing high protein levels in their seeds (rather than throughout the plant as in duckweeds). Both soybean and duckweed have a proclivity for taking up ammonium as well as accepting ammonium from their endogenous N_2_-fixing *Rhizobium* symbionts. 

## 6. Interactions among Nutrient Transporters, Hormones, Elevated CO_2_, and the Microbiome

This section highlights interactions between elevated CO_2_ and transport/metabolism of nitrogen compounds while comparing the response of duckweeds with that of various terrestrial plants. Duckweeds can form thin, slender root-like appendices and increase their length, and thus the plant’s surface-to-volume area, for nutrient uptake under low nitrogen supply [70]. “Nitrate and hormonal signaling crosstalk” (for a review, see Vega et al. [77]) has been characterized in *Arabidopsis thaliana*, where a high-affinity nitrate transporter was linked to regulation of lateral root growth and increased access to mineral nutrients under limiting nutrient supply [77]. Elevated CO_2_ has been shown to alter the expression of transporters for nitrogen compounds in a manner that depends on nutrient concentration, plant species, and overall plant response to elevated CO_2_ [78]. In tall fescue, elevated CO_2_ increased root growth and did so even more under moderate versus low nitrate supply [79]. In other scenarios, elevated CO_2_ decreased, and thus interfered with, high-affinity uptake of nitrogen compounds under limiting nitrogen supply [78]. The plant microbiome can stimulate root growth and nutrient uptake in various species [80], including rice [81,82], maize [83], and soybean [82]. Duckweeds [84] and soybean [85] can thrive under low nitrogen supply due to partnerships with N_2_-fixing endophytic microbial symbionts, such as *Rhizobium lemnae* for *Lemna* species [86] and various rhizobia species for soybean [87]. The ability to metabolize large amounts of ammonium thus presumably allows duckweeds to take advantage of plant-microbe interaction to restore carbon-nitrogen balance under elevated CO_2_ in combination with low nitrogen supply (see Figure 3 above). In such situations, the microbiome (i) acts as a carbon sink that helps remove surplus sugar, (ii) supports nitrogen use for new growth, and (iii) supports a plant hormone balance conducive to continued new growth, all of which counteract feedback downregulation of the photosynthetic apparatus and the associated aspects of plant nutritional quality. 

Whether or not elevated CO_2_ stimulates plant productivity can also depend on interaction between the plant microbiome and plant developmental stage. A stimulation of soybean seed yield by elevated CO_2_ was only seen in nodulated soybean lines (with endophytic N_2_-fixing symbionts) but not in non-nodulated mutant lines of soybean and only during the reproductive phase—with no growth stimulation during the vegetative phase of plant growth [88]. Developmental effects should play a minor role in duckweeds that mainly reproduce by vegetative cloning and rarely flower [89]. No additional improvement in soybean productivity was obtained when using bacteria with higher N_2_-fixation rates for inoculation [90]. Moreover, inoculation with bacterial strains isolated from soybeans grown under elevated CO_2_ caused metabolic disruption in plants grown under ambient CO_2_ [91].

## 7. Commercial Applications of Duckweed

### 7.1. Human and Livestock Nutrition

Based on its nutritional composition, duckweed can provide adequate nutrition for both humans and livestock. Due to accumulation of large amounts of vegetative storage protein throughout the plant, duckweed can produce between 20 and 30% of crude protein, which is equivalent to 20× the amount of protein produced by soybeans (per unit cultivated area) [92] and even much higher amounts than those produced by cereals [93]. A key factor in the superior protein production by duckweed is protein accumulation in all its tissues compared to only in the seeds of land plants. Furthermore, duckweed protein quality is the same as that of soybean or milk—with all essential amino acids humans must obtain from their diet [94]. The quantity and quality (amino acid profile) of crude protein are directly related to the amount of nitrogen in the medium (see Section 5) [95]. The main protein in duckweed is RuBisCO, a non-allergenic source of all nine essential amino acids with high digestibility, making it an ideal candidate for animal feed [96]. Duckweed also contains large amounts of the non-essential amino acid glutamic acid, which has health benefits for humans and other animals [97]. However, similar to other plants (e.g., spinach or beet greens [98]), some duckweeds (e.g., genus *Lemna*) can contain high levels of anti-nutritive compounds such as oxalic acid [99]. This compound chelates calcium ions and other micronutrients, hindering their absorption in the intestine and compromising human health [100,101]. In contrast, duckweeds in the genus *Wolffia* contain minimal and thus safe amounts of anti-nutrients, which makes this genus suitable as a safe source of human food and highly recommended due to its superior nutritional characteristics regarding proteins, vitamins, and other micronutrients mentioned above. There are other potentially anti-nutritive compounds (nitrates, tannins, phytates, dioxins, and cyanogenic glycosides [102]) that require further investigation in duckweeds. *Wolffia globosa* (Mankai) has long been consumed as a high-protein source in Japan [103].

Sufficient intake of protein and other micronutrients (such as vitamins and antioxidants) is essential for optimal body function [50] and one of the current global challenges [104] facing human society and science. Currently, more than 15% of people worldwide cannot achieve the recommended daily protein intake and fall behind even further with respect to micronutrients (up to a 25% deficit), leading to malnutrition [105,106]. The WHO “Global Strategy for Food Safety (2022–2030)” program (https://www.who.int/publications/i/item/9789240057685; URL accessed on 27 August 2023) aims to “ensure that all people, everywhere, consume safe and healthy food so as to reduce the burden of foodborne diseases” [107]. A resilient food system is thus needed that is fit for the future, with concomitant benefits for the health of individual people and societies as well as for the physical and natural environment, including climate.

Plant–microbe interaction in duckweed also has additional nutritional benefits beyond minimizing the loss of photosynthesis-associated micronutrients under elevated CO_2_ (see above; [39,40]). In contrast to edible products from other plants, Mankai contains high levels of vitamin B12—even after plants have been surface sterilized [108]. The synthesis of vitamin B12 is carried out by bacteria and archaea absent in other vegetable food products [109]. Presence of vitamin B12 in Mankai is thus apparently due to as-yet undefined B12-producing bacteria that must be endophytic and produce high levels of vitamin B12 even in otherwise axenic Mankai cultures [94,108]. Because other plants are not a source of vitamin B12, this vitamin is currently obtained mainly from animal products (clams and beef liver are rich in vitamin B12; [110]). The current trend towards greater consumption of plant versus animal products [111] increases the risk of deficiencies in vitamin B12 and associated health problems [112]. To meet the need for B12 intake in diets low in animal products, algae had been suggested as a source for this compound, but recent studies showed that vitamin B12 occurs in algae in a form (pseudo B12) not usable by humans and even capable of competing with the usable forms of B12 [113]. In contrast, Makai can be considered a good addition to meat-free diets. However, the benefits of Mankai go farther because Mankai intake also positively modified the gut microbiota and increased the diversity of genes (as assessed via 16S rRNA) related to the absorption of vitamin B12 [108]. A recent series of human clinical trials provided a comprehensive summary of these health benefits, including improved serum levels of vitamin B12, iron, and phenolics, gut microbiome composition, brain function, blood glucose control/insulin sensitivity, cardiovascular health, weight management, and decreased fat accumulation resulting from supplementation with a combination of Mankai and green tea [108,114,115,116,117,118,119,120,121].

### 7.2. Feedstock for Biofuels

This section highlights plant use as feedstock for biofuels while comparing relevant features of Lemnaceae and terrestrial plants. Problems with current production systems for biofuel feedstocks include food–fuel conflict and undesirable environmental impacts, both due to the use of staple food crops with relatively high requirements for irrigation and fertilization. Production of maize, yucca, or sweet potato for biofuel increased the price of basic foods and threatened food security [122]. In association with these crops’ high fertilizer requirement, excessive nutrient runoff caused environmental problems [123]. Therefore, it is urgent to identify new, sustainable feedstocks for biofuel production.

Due to its high starch content (see, e.g., [38]), duckweed is suitable for bioethanol production via starch hydrolysis and subsequent alcoholic fermentation. The yield of bioethanol from duckweed has been estimated to potentially reach 6420 L ha^−1^, which is about 50% above that obtained for ethanol based on corn [3,123,124] and production could continue throughout the year. The ability of *Lemna* to accumulate starch without needing large amounts of light [38] and to respond to elevated CO_2_ with increased starch accumulation would support high biofuel production yields [4]. Biofuel production could thus be paired with wastewater treatment, as previously shown for systems that combine wastewater recycling with livestock feed production [92]. Moreover, effective sequestration of atmospheric CO_2_ into starch by duckweed would contribute to CO_2_ mitigation and climate control. Such systems could serve as a model for future regenerative life support systems, providing both food and fuel.

Additional mechanisms have been documented through which starch accumulation by *Lemna* can be enhanced through restriction of nutrient supply with minimal impact on growth rate [123,125,126]. Restriction of sulfur supply showed the greatest promise with respect to stimulation of starch accumulation with negligible growth impact [125], while nitrogen limitation increased starch yield at the expense of growth and overall biomass production [125]. Recent progress has also been reported in engineering duckweed for enhanced accumulation of lipids [127] that could be used to produce biodiesel.

## 8. Agricultural Technology: Hydroponics and Vertical Farming

This section highlights plant use in vertical and hydroponic farming while again comparing relevant features of Lemnaceae and terrestrial plants. Documented advantages of hydroponic crops include the highly efficient use of water and nutrients [6] by crops such as tomatoes, cucumbers, and other vegetables [6,72]. Recirculating hydroponic systems further increase resource-use efficiency [128]. Duckweed has a high potential for use in closed hydroponic systems consisting of multiple vertical layers (Figure 6), with controlled temperature, CO_2_, light intensity, quality, and photoperiod, as well as nutrient composition [129]. Vertical design would further optimize use of space [72,105,130] when using a diminutive, 100% edible plant like duckweed. Control of environmental factors allows optimization of biomass quality, i.e., micronutrients (vitamins, antioxidants) and proteins for use as food [131] or either starch or lipid content for use as biofuel feedstock. Recirculation of nutrients and water prevents nutrients release into the environment and resulting land and water pollution [132], as well as minimizes water loss [128] by reducing evaporation (proximity between different vertical levels could act as “semi-closed lids”). Duckweed, furthermore, thrives on a film of water [133]. Vertical, hydroponics-based farming is also suitable for continuous production throughout the year. 

Figure 6 depicts a system where layers of duckweed are interspersed with layers of light-emitting diodes (LEDs), resulting in illumination of duckweed layers from both below and above, a design first suggested by Gale et al. [48]. Figure 6 features a mix of blue and red LEDs that, due to minimal self-shading within a single layer of fronds, may support maximal productivity with minimal energy input. We suggest using both red and blue wavelengths to support not only photosynthesis but also plant morphogenesis [134] and blue-light-triggered phenolic production [135].

## 9. Conclusions

Duckweed is an attractive candidate to sequester atmospheric CO_2_ while at the same time producing multiple products and services of human interest. In addition, the aquatic cultivation of *Lemna* avoids releasing CO_2_ into the atmosphere from the soil, as can occur during terrestrial plant cultivation. 

Due to its high lipid- and starch-storage capacity, duckweed is a promising crop for production of biofuels, and its exceptionally high protein and human micronutrient content makes it a food with high nutritional value. Duckweed’s outstanding ability to absorb ammonium (and turn it into protein), furthermore, serves in wastewater treatment. Its ability to sequester CO_2_ is enhanced by presence of a microbiome that counteracts the accumulation of surplus sugars and resulting feedback inhibition of photosynthesis (and associated loss of human micronutrients).

All these characteristics make duckweed unusually productive in small cultivation areas or volumes. The combination of its diminutive size with exceptionally high relative growth rates, harvest index (no inedible parts), and nutrient density makes this plant ideal for vertical growth in multiple stacked layers (Figure 6). Whereas multi-layer cultivation may also be attractive with terrestrial plants, it offers clear advantages in duckweed with highly efficient use of space, energy, nutrient supply, and gas exchange with the atmosphere.

## Figures and Tables

**Figure 1 plants-12-03090-f001:**
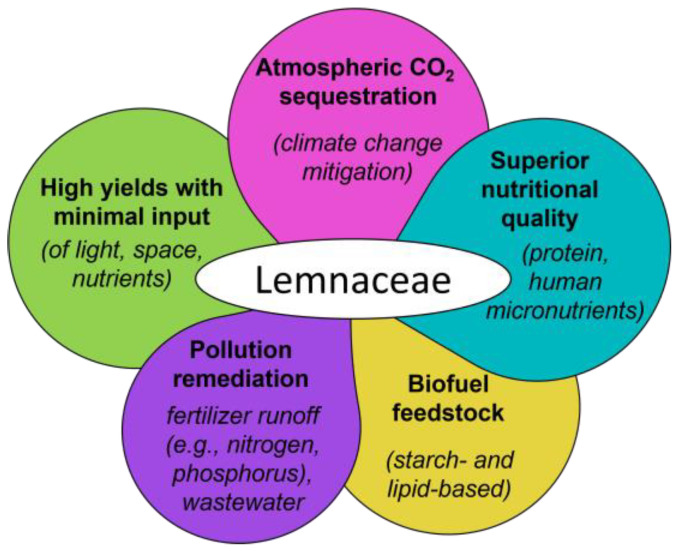
Schematic summary of the many applications of duckweed coupled with its high resource-use efficiency. Further detail is addressed in the different sections of this review.

**Figure 2 plants-12-03090-f002:**
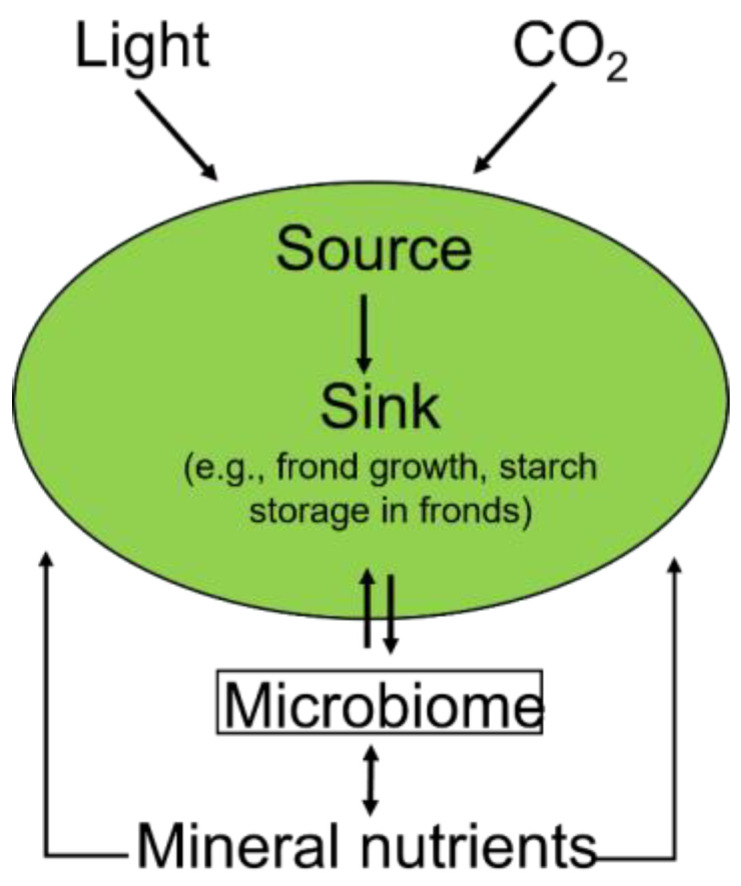
Schematic depiction of duckweed fronds serving as both source and main sink for photosynthetically produced sugar and the bidirectional interaction of the plant with its microbiome serving as a sugar sink and assisting with nutrient acquisition by increasing bioavailability and uptake by the plant. By modulating plant hormones, the microbiome may be able to modulate expression of high-affinity nitrate and/or ammonium transporters (that are also tied to regulation by phytohormones in plant roots).

**Figure 3 plants-12-03090-f003:**
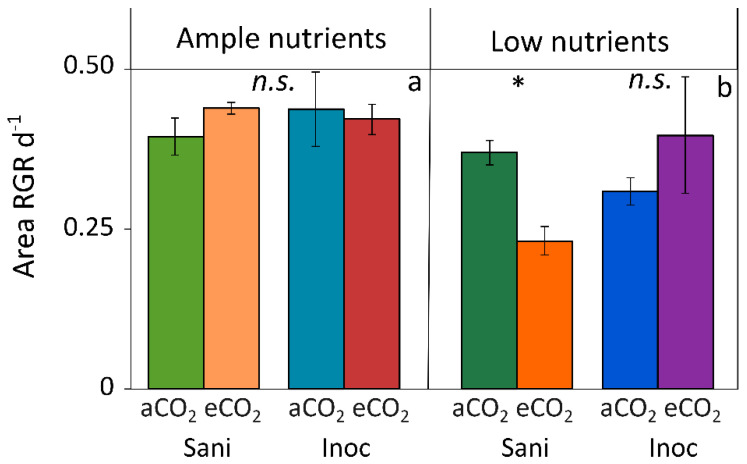
Relative growth rate (RGR; based on frond area expansion per day) of *L. minor* growing under either (**a**) ample nutrients (1/2 strength Schenk & Hildebrandt medium; [44]) with ambient (aCO_2_) or elevated (eCO_2_) CO_2_ and without (Sani) or with (Inoc) its microbiome or (**b**) low nutrients (1/20 strength Schenk & Hildebrandt medium) with ambient or elevated CO_2_ and without or with its microbiome. Mean values ± standard deviations, *n* = 3. * = significantly different at *p* < 0.05 (Student’s *t*-test); *n*.*s*. = not significantly different. Original data from Zenir et al. [39].

**Figure 4 plants-12-03090-f004:**
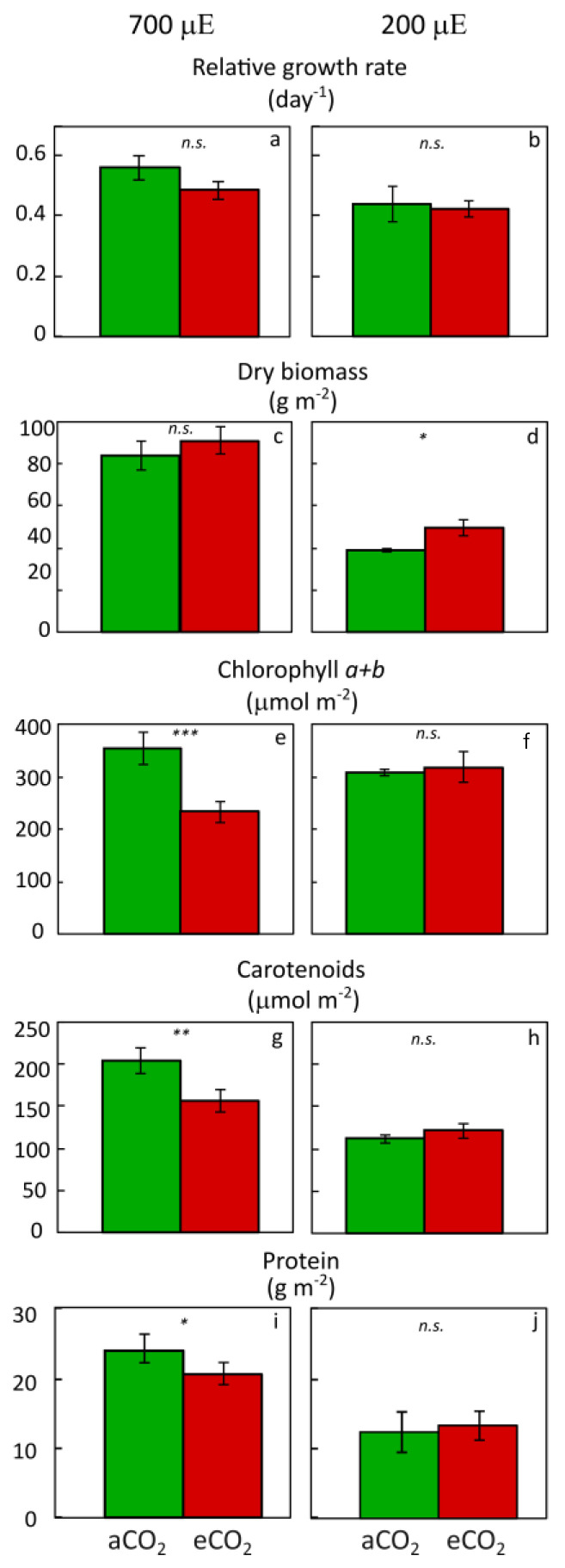
(**a**,**b**) Relative growth rate (RGR), (**c**,**d**) dry biomass, (**e**,**f**) and chlorophyll (**g**,**h**), carotenoid, and (**i**,**j**) protein content, all on a frond area basis, of *L. minor* growing under two light levels with ample nutrient supply and in the presence of either ambient (aCO_2_) or elevated (eCO_2_) CO_2_. Light supply was either very high, with 700 µmol photons m^−2^ s^−1^ (700 µE) for 24 h per day, or lower, with 200 µmol photons m^−2^ s^−1^ (200 µE) for 24 h per day. All plants were associated with their microbiomes. Ample nutrients were supplied as 1/2 strength Schenk & Hildebrandt medium [44]. Mean values ± standard deviations, *n* = 3. All paired sets of data were subjected to a Student’s *t*-test, with asterisks indicating the level of significant differences: *n*.*s*. = not significantly different; * = *p* < 0.05; ** = *p* < 0.01; *** = *p* < 0.001. For original data, growth conditions, and methods, see Demmig-Adams et al. [37] and Zenir et al. [39]. Previously unpublished data in panel **j** are from Marina López-Pozo.

**Figure 5 plants-12-03090-f005:**
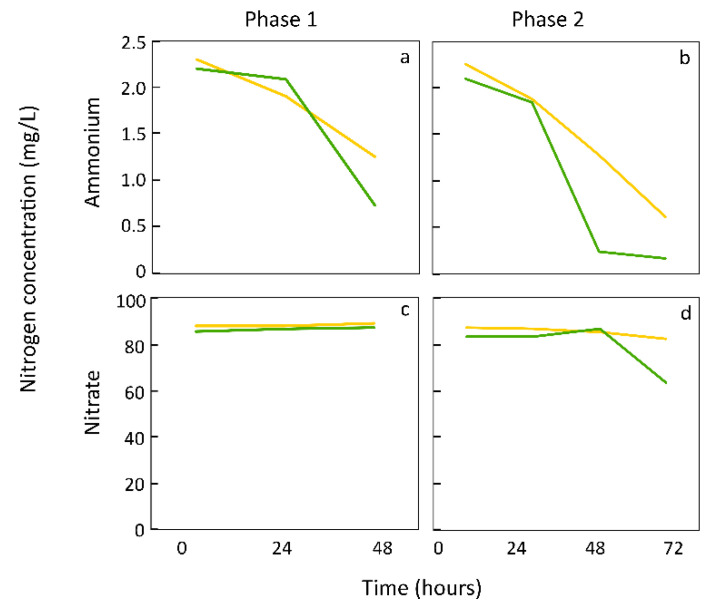
Ammonium (**a**,**b**) and nitrate (**c**,**d**) uptake by sanitized *L. minor* during phase 1 (48-h growing phase upon transfer of four fronds from ½ to 1/20 Schenk & Hildebrandt medium) and phase 2 (72-h second growing phase in 1/20 Schenk & Hildebrandt medium after transfer of fronds to fresh medium at time 0 of phase 2 and concomitantly reducing the number of fronds to four) under ambient CO_2_. This was performed under either a short (12 h light/12 h dark) photoperiod (green lines) or a long photoperiod (24 h light; yellow lines) with 200 µmol photons m^−2^ s^−1^ in each case. Ammonium and nitrate concentrations in the media were determined photometrically (photometers HI83314 and HI83300, respectively; Hanna Instruments, Woonsocket, RI, USA). At the time of sampling, 10 mL of media were collected for ammonium determination and an additional 1 mL for nitrate. *n* = 3 for each time point. Unpublished data from Madeleine Zenir and Marina López-Pozo. Plants were grown as described in Zenir et al. [39].

**Figure 6 plants-12-03090-f006:**
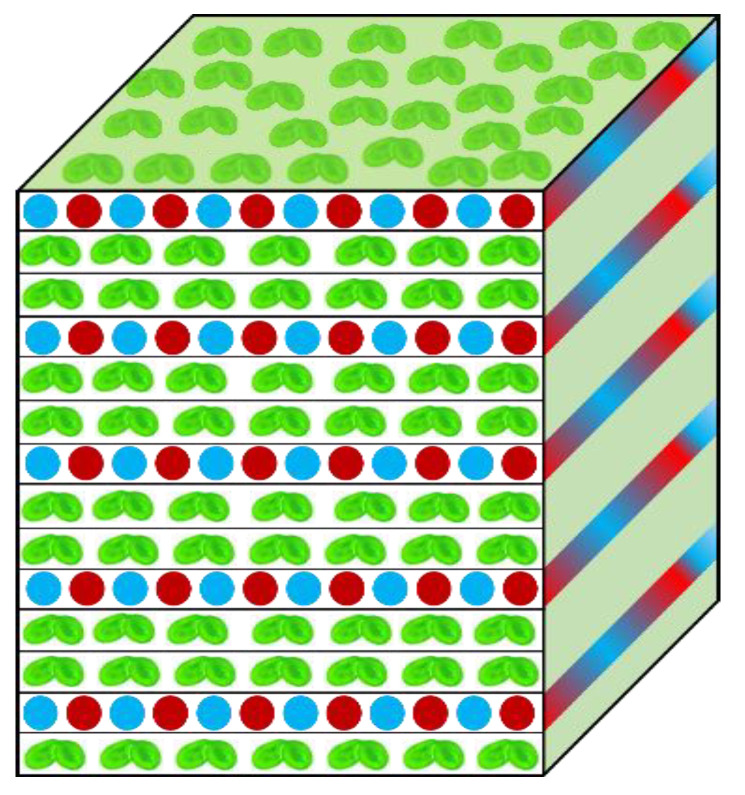
Schematic representation of a multi-level, closed, vertical production system for duckweed with interspersed blue and red light emitting diodes supplying the low level of light that is sufficient to saturate plant growth and food or biofuel feedstock production.

## Data Availability

The data discussed in this review are shown here and/or in previously published studies.

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
