# Peer review of "Lemnaceae as Novel Crop Candidates for CO_2_ Sequestration and Additional Applications"

_plants, 2023, doi:10.3390/plants12173090_

Round 1

Reviewer 1 Report

The authors present a fascinating manuscript reviewing the response of duckweed (mainly Lemna minor) under different environmental conditions including different light intensities, sterile and nonsterile plants, different levels of nutrient (based on SH medium), different ratios of nitrate and ammonium. New insights were obtained concerning the influence of elevated CO2 on protein content and carotenoids. The manuscript is of general interest for nutrition and industrial applications.

The main problem in this manuscript is the fact that the authors careless transfer general physiological knowledge of terrestrial plants to water plants, i.e. duckweeds. This is especially evident in chapter 1.2 and Fig. 2 but basically throughout the whole manuscript. Large parts of the manuscript gain this way the character of a speculation. See line 116/117: “More research is needed to elucidate such responses in duckweed” could stand for several parts. Even worse, this become not always clear to the reader without consulting the cited references.

Therefore, I suggest major revision of the whole manuscript making clear what might be hoped for duckweeds on the basis of general knowledge of physiology of plants. Nevertheless, the manuscript is highly interesting and I suggest to publish it in PLANTS after thoroughly carried our revision.

I suggest to consider one of the papers of T. Pfannschmidt concerning photosynthetic acclimation (e.g. Nature 437: 1179 (2005) or Hommel et al. Frontiers in Plant Science 12, 2022) related to chapter 4 of the present manuscript.

Reviewer 2 Report

The paper deals with important climate change mitigation and carbon sequestration subjects. Lemnaceae was analysed as a Novel Crop Candidate for CO2 Sequestration. The paper is well-written however paper does not follow IMRAD profile. It misses discussion section and literature review is pour. The results of this paper need to be discussed and compared in the light of other studies so show the input of this paper. There are also no conclusions. The only sentence in the end this is not real conclusions providing the main findings and implications of conducted research.  Paper needs major revision.

Reviewer 3 Report

see doc

Round 2

Reviewer 1 Report

The authors dramatically revised the original manuscript considering now also relevant duckweed publications. This should have been done already in the first version. I suggest to publish the revised manscript in its present form.

Reviewer 2 Report

The authors did big job and revised they paper in proper way. My all comments were addressed in general and relevant answers were provided to my comments. I do not have more comments and I think that paper can be publushed in current form.